# Muscle growth and anabolism in intensive care survivors (GAINS 2.0): Protocol for a multi-centre randomised; placebo controlled clinical trial of nandrolone in deconditioned adults recovering from critical illness

**Matthew Anstey**[1,2,3]*, **Ed Litton**[3,4], **Maryam Habibi**[3], **Lisa Van der Lee**[1], **Robert Palmer**[1], **Natalie Tran**[1], **Bianca Mammana**[1], **Stacey Scheepers**[1], **Annamaria Palermo**[4], **Xavier Fiorilla**[1], **Bhaumik Mevavala**[1], **Adrian Regli**[5], **Angela Jacques**[6], **Bradley Wibrow**[1,3]

1 Intensive Care Department, Sir Charles Gairdner Hospital, Perth, Western Australia, Australia, 2 School of Population Health, Faculty of Health Sciences, Curtin University, Perth, Western Australia, Australia, 3 School of Medicine, University of Western Australia, Perth, Western Australia, Australia, 4 Intensive Care Department, Fiona Stanley Hospital, Perth, Western Australia, Australia, 5 Intensive Care Department, St John of God Hospital Murdoch, Perth, Western Australia, Australia, 6 Institute for Health Research, University of Notre Dame, Fremantle, Western Australia, Australia

* Matthew.anstey@health.wa.gov.au

**Data Availability Statement:** No datasets were generated or analysed during the current study. All

## Abstract

### Background

Intensive care patients can experience significant long-term impairment in mobility and function caused by their critical illness. A potential contributory factor apart from critical illness polymyoneuropathy is the low levels of anabolic hormones in these patients. Testosterone levels in critically ill patients are extremely low, even in the latter recovery phase. A potential solution to critical illness myopathy may be to provide anabolic support in addition to standard care (early physiotherapy) to further improve gains in strength.

### Research question

This project aims to test whether a synthetic testosterone (nandrolone) improves muscle strength in ICU survivors compared to placebo.

### Methods

GAINS 2.0 is a multicentre, randomised, double blinded placebo-controlled trial which will allocate ICU patients in a 1:1 ratio to nandrolone compared to placebo which commenced recruitment in July 2023. Adult patients admitted to the ICU, receiving nutrition for a minimum of 24 hours with an ICU stay of at least 5 days, or patients with significant weakness as result of their ICU stay (such that they are unable to mobilise independently) will be eligible to participate. Sample size will be 54 patients. Patients will be randomised to receive nandrolone 100mg (males) / 50mg(females) weekly for 3 weeks in addition to standard care.

relevant data from this study will be made available upon study completion.

**Funding:** The author(s) received no specific funding for this work.

**Competing interests:** The authors have declared that no competing interests exist.

**Abbreviations:** ANZROD, The Australian and New Zealand Risk of Death; APACHE, Acute Physiology and Chronic Health Evaluation; BMI, body mass index; CKS, Chronic kidney disease; CPA, Chelsea critical care physical assessment; CRP, C-Reactive protein; CV, Cardio vascular; DSMB, Data and Safety monitoring board; EF, ejection fraction; FDA, Food and Drug Administration; HDU, high dependency unit; ICU, intensive care unit; ICUAW, ICU associated weakness; IMI, Intramuscular injection; LFT, Liver function test; LOS, Length of stay; MRC, Medical Research Council; QALY, Quality-adjusted life year; REDcap, Research Electronic Data Capture; SAE, Severe adverse events; SBP, Systolic blood pressure; SF-36, Short form health survey; STEMI, ST-elevation myocardial infarction; WBC, White blood cell.

The co-primary outcomes are the time to walking with one person assisting (Intensive Care Mobility scale = 8 or more, in days from randomisation), change in muscle strength measured by the Medical Research Council (MRC) muscle strength sum score from enrolment to hospital discharge and number of days out of hospital up to day 90 post-discharge. Secondary outcomes are grip strength measured by hand-held dynamometry. SF-36 scores (quality of life and functional domains), and days to return to work, for those working pre-ICU, will be collected via a 3-month phone follow-up.

## Conclusions

A previous pilot feasibility trial showed that nandrolone is safe and feasible. We hypothesize nandrolone will improve muscle strength and physical functioning at hospital discharge and at follow-up. The results of this trial may have significant interest to clinicians and patients considering the large and increasing number of patients surviving intensive care but with physical impairment. This trial may have significant implications on lowering hospital costs and daily adjusted life years.

## Trial registry

anzctr.org.au; No.: ACTRN12623000729628 URL: anzctr.org.au.

## 1. Background

Despite recent advances it is estimated that one-quarter to one-half of long-stay intensive care survivors live with significant weakness as a consequence of their illness, resulting in impaired mobility and function [1, 2]. This has flow on effects in terms of returning to previous levels of functioning and work [3]. The loss of muscle mass in critical illness is related to immobility and a complicated process that causes muscle and nerve dysfunction called critical polymyoneuropathy [4, 5]. Early physiotherapy and early mobilisation [6] can improve muscle synthesis and patient outcome [7]. Nonetheless, muscle protein catabolism during critical illness is an independent contributor to longer hospital stay [8–10]. A contributory factor may be reduced levels of anabolic (muscle building) hormones such as testosterone. Testosterone levels have been reported to be low in critically ill patients with COVID [11], chronic kidney disease [12] and ICU patients even in the recovery phase from acute illness [8, 10]. One potential treatment may be to provide anabolic support in the recovery phase from prolonged critical illness.

The best studied anabolic hormone is testosterone. Testosterone has been studied in heart failure with success in improving functional capacity and cardiovascular risk factors [13, 14]. Studies in critically ill burns patients show that testosterone treatment reduced weight loss and length of stay (LOS) whilst improving donor site healing time. Previous treatment studies with oxandrolone supplementation have shown mixed results [15–19]. However, nandrolone is a synthetic myotrophic testosterone with a good safety and efficacy profile [20–23].

Direct studies assessing the effect of testosterone on muscle anabolism have been published on septic rat models with improvement in contractile force [24]. The synthetic anabolic steroids nandrolone or oxandrolone exhibit significantly greater selectivity for myotropic properties, with minimal androgenic effects, potentially minimising any adverse outcomes [14]. Nandrolone has previously been used successfully (compared with testosterone or placebo) in

a randomized controlled trial to reverse weight loss in human immunodeficiency virus/ acquired immunodeficiency syndrome patients and in patients with chronic obstructive pulmonary disease to improve respiratory function and muscle wasting [20–22]. Recent retrospective analysis and case series with limited number of patients have shown improvement in grip strength, ambulation and functional recovery in patients with ICU associated weakness (ICUAW) [25, 26]. In the critical care setting, the authors have previously conducted a pilot feasibility trial of nandrolone versus placebo, demonstrating the feasibility and safety of nandrolone [27]. The study showed reduction in hospital LOS and mechanical ventilation [27]. Grip strength, mobility and function were not significantly changed, but the pilot was underpowered to assess these outcomes, highlighting the need for larger randomised controlled trials.

The primary objective of this trial is to determine whether the addition of nandrolone to standard care, when given to adult patients with critical illness myopathy in the recovery phase from acute illness, reduces hospital length of stay and reduces the time to the patient to first achieving the milestone of walking away from the bedside with one person assisting, without the use of a walking aid.

## 2. Methods

### 2.1. Design

This study is a multicenter double blinded randomised, placebo-controlled intervention trial conducted in Intensive Care Units (ICU) in several hospitals across Western Australia. ICU patients will be allocated to a 1:1 ratio of nandrolone in addition to standard care compared with a placebo in addition to standard care. Nandrolone or placebo will be administered intramuscularly (IM) weekly for up to 3 weeks. Standard care includes physiotherapy sessions (including early mobilisation), and nutritional support. Placebo is defined as sterile water or normal saline injection. Fig 1 outlines timepoints of enrolment, interventions and outcomes. An outline of the study design is provided in Fig 2 in accordance with CONSORT reporting guidelines [28]. Recruitment commenced on 5[th] July 2023.

### 2.2. Study population

Adult patients admitted to the ICU or HDU with minimum of 5 days OR patients who experience significant weakness below their baseline level (see inclusion criteria) as a result of their ICU illness/stay will be considered for the study. A total of 60 patients will be recruited in this trial.

### 2.3. Eligibility criteria

**2.3.1. Inclusion criteria.** Patients who are 21 years of age, admitted to participating ICU/ HDU with a LOS minimum of five days or who experience weakness below their baseline functioning (practically meaning their mobility is worse than the primary outcome of being able to walk with the assistance of one person) because of their critical illness and their ICU stay. They need to be receiving nutrition at estimated goals for at least 24 hours. Table 1 outlines the exclusion criteria.

### 2.4. Interventions

Eligible patients will be randomised to receive nandrolone 100 mg (males)/50mg (females) or placebo weekly for 3 weeks or until hospital discharge, whichever is earlier.

| TIMEPOINT | Enrolment | Allocation | Post-allocation | | | | | | Close-out |
|---|---|---|---|---|---|---|---|---|---|
| | **-t₁** | **0** | **D 7** | **D 15** | **D 21** | **Hospital discharge** | **3/12** | **6/12** | **t_x** |
| **ENROLMENT:** | | | | | | | | | |
| **Eligibility screen** | X | | | | | | | | |
| **Informed consent** | X | | | | | | | | |
| **Demographic information** | X | | | | | | | | |
| **Allocation** | | X | | | | | | | |
| **INTERVENTIONS:** | | | | | | | | | |
| *Nandrolone* | | ◆———————————◆ | | | | | | | |
| *Standard care* | | X | X | X | X | X | | | |
| *Therapies received – physiotherapy duration and nutrition (calories)* | | X | X | X | X | X | | | |
| **ASSESSMENTS:** | | | | | | | | | |
| *BASELINE:* Baseline sex hormones, blood tests (liver function, lipids, handgrip strength, weight, medical research council (MRC)l sum score strength, intensive care mobility scale | X | X | | | | | | | |
| *OUTCOMES:* Intensive care mobility scale, grip strength, MRC score, hospital length of stay | | | X | X | X | X | | | |
| **Mortality** | | | | | | X | X | X | |
| **SF-36 score** | | | | | | | X | X | |
| **Return to work (for working age enrolments)** | | | | | | | X | X | |

**Fig 1. SPIRIT figure outline schedule of interventions and outcome measures.**

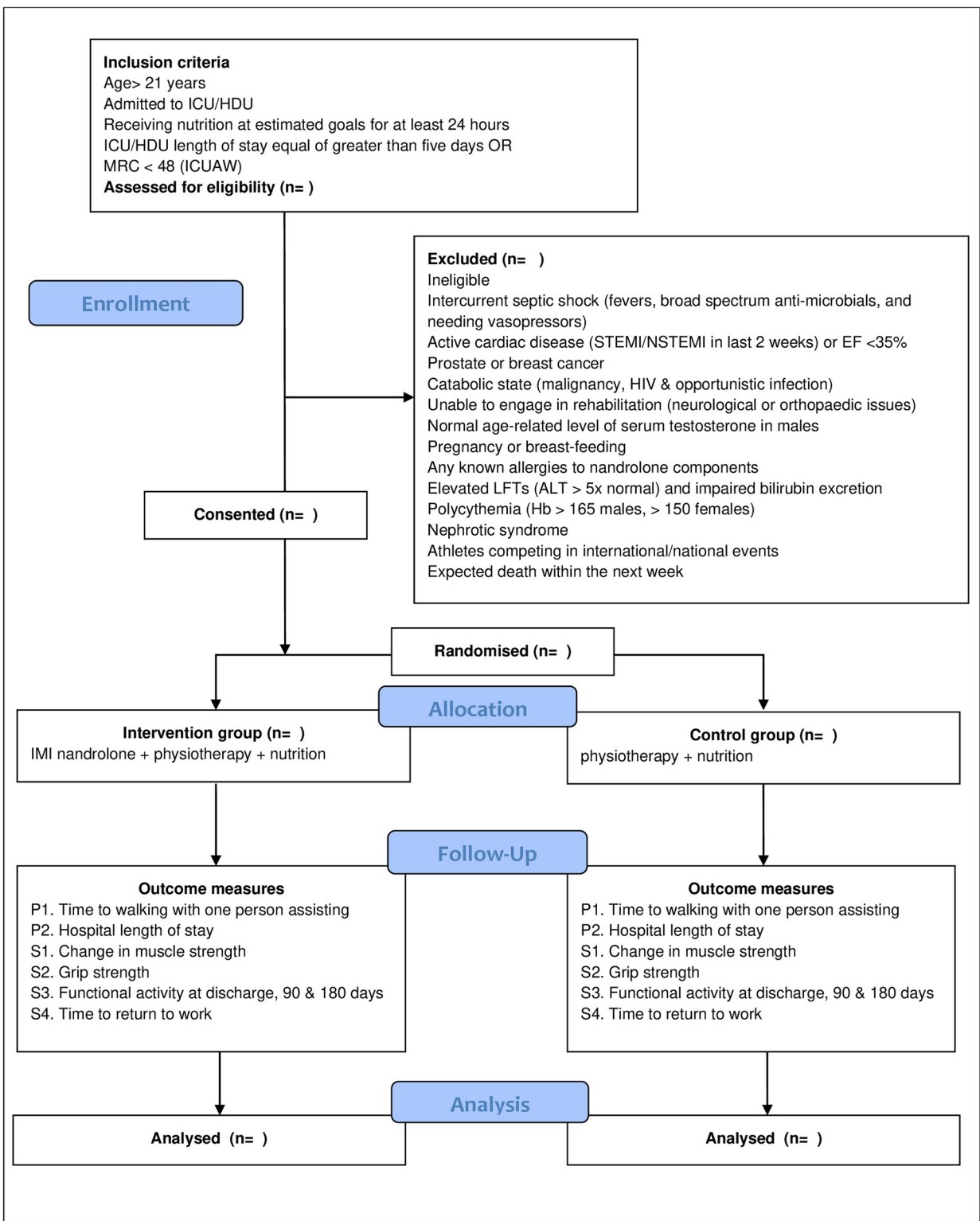

**Fig 2. Outline of study design.**

**Table 1. Exclusion criteria.**

| Exclusion criteria | Justification |
|---|---|
| Intercurrent septic shock (fevers, broad spectrum anti-microbials, and needing vasopressors) | Therapy designed for recovery phase after acute illness subsided |
| Active cardiac disease (such as STEMI/NSTEMI in last 2 weeks) or EF <35% | Testosterone has potential to worsen heart failure [29]) |
| Prostate or breast cancer | Contraindication |
| Ongoing reason for catabolic state (active malignancy, HIV & opportunistic infection last 2 months) | Confounding factor |
| Unable to engage in rehabilitation (due to significant neurological or orthopaedic issues, included spinal cord injury/damage) | Prevents rehabilitation |
| Normal age-related level of serum testosterone in males (measured in early morning 6-9am) | Contrary to rationale of therapy |
| Pregnancy or breast-feeding | No safety data available |
| Any known allergies to nandrolone components—including peanuts and soya and latex | Contraindication |
| Elevated LFTs (ALT > 5x normal) | Contraindication |
| Polycythaemia (Hb > 165 males, > 150 females) | Contraindication |
| Current nephrotic syndrome | Contraindication |
| Athletes competing in international/national events | Prevents future competition |
| Expected death within the next week | Not going to reach outcomes of interest |

The study drug will be drawn up by a blinded member of the investigational team not otherwise involved in the care of the participant (e.g. pharmacist or senior nurse) with the syringe covered (to mask the contents of the syringe). Both groups (intervention and control group) will receive concomitant standard therapies including physiotherapy (duration of rehabilitation and the highest mobility level recorded) and nutritional assessment and support (caloric intake, protein intake, food intake and extrapolated calories and protein based on meal charts from hospital to be recorded) in addition to medical interventions as per treating physician.

## 2.5. Blinding and randomisation

A local investigator will screen ICU patients daily. All investigators, clinicians, nursing staff and patients are blinded except for the ICU pharmacist or senior nurse (not directly involved in patient care) who draws up the drug. Syringes will be masked to prevent the contents being visualised to the administering nurse. Eligible patients ready to be assigned to study treatment will be randomised via a permuted block randomisation with variable block sizes in a 1:1 ratio with site stratification. Allocation sequence will be provided via an independent external source.

## 2.6. Outcomes

The co-primary outcomes are the time to walking with one person assisting (Intensive Care Mobility scale = 8 or more, in days from randomisation), change in muscle strength measured by the Medical Research Council (MRC) muscle strength sum score from enrolment to hospital discharge and number of days out of hospital up to day 90 post-discharge. Secondary outcomes are grip strength measured by hand-held dynamometry. SF-36 scores (quality of life and functional domains), and days to return to work, for those working pre-ICU, will be collected via a 3-month phone follow-up. Economic evaluation in the form of cost consequence performed by a medical economist will determine any change in ICU and hospital stay.

Additionally, QALYs will also be estimated using the SF-36 data converted to SF-6D using Australian population weights [30].

## 2.7. Statistical analysis

**2.7.1. Power calculation.** Power calculations for primary outcomes MRC (continuous), IMS>8 (binary) and number of days out of hospital since enrolment out of 90 days (count) and secondary outcome grip strength follow. A sample of n = 46 has 80% power (alpha = 0.05) to detect a mean±SD difference of 7±10 in MRC and grip strength (effect size difference f = 0.35) between 2 groups over a minimum of 3 timepoints in a repeated measures ANOVA between factors (mixed) model. A sample of n = 46 has 80% power (alpha = 0.05) to detect an odds ratio of 2.75 in a logistic model, based on a group proportional difference of 24.5% for IMS>8 (pilot study indicated a 27.2% proportional difference). Based on a Poisson regression for count data, a sample of n = 44 has 80% power to detect a group difference in number of days out of hospital at 90 days from enrolment of 7 (70 vs 63 days). Allowing for 15% loss to follow-up, a total of 54 patients will be recruited. G*Power 3.1.9.7 (University of Dusseldorp and PASS Sample Size (NCSS, LLC) was used for power calculations.

**2.7.2. Statistical methods.** The analysis will be by intention to treat population, defined as all study participants except for those who do not consent to the use of data.

Descriptive summaries of patient and outcome data will consist of means and standard deviations or medians and interquartile ranges for continuous data and frequency distributions for categorical data. Univariate group comparisons between treatment arms will be done using t-tests or Mann-Whitney U tests for continuous data, depending on normality of distributions, and Chi-squared or Fisher's Exact tests for categorical data comparisons.

Generalised linear mixed models (GLMM) will be used to examine longitudinal MRC and grip strength with results summarised as estimated means and mean differences with 95% confidence intervals (CI). Interaction effects (difference of the group differences over time) will be reported. Mixed effects logistic models will be used to examine the odds of walking independently (IMS>8) with results summarised as odds ratios (OR) and 95% CI.

For surviving patients, Poisson regression will be used to compare number of days out of hospital at 90 days and days to return to work from enrolment between groups, with results summarised as estimated means and mean differences with 95% CI.

Kaplan-Meier survival probabilities and Cox regression models, accounting for the competing risk of mortality and hospital readmission, and censoring at 90 days, will be used to compare number of days out of hospital up to 90 days post-discharge and days to return to work from enrolment between groups, with results summarised respectively as median number of days with 95% CI and cause-specific hazard ratios (HR) with 95% CI.

Linear regression models will be used to compare SF36 quality of life subscale outcomes between groups, with results summarised as estimated means and mean differences with 95% CI.

All mixed effects models will incorporate random subject effects and group-time interactions. Unadjusted models for non-primary outcomes will also be compared to adjusted models that separately include covariates ICU length of stay, age, gender, BMI, APACHE 2 score and baseline MRC to examine covariate effects and whether inclusion is advised.

Times to mortality and morbidity events will be described using Kaplan-Meier probabilities and compared between groups using log-rank tests. Results will be summarised using medians and 95% CI. Cox proportional hazards regression models may be used to describe adjusted hazard ratios (HR) and 95% CI where univariate differences are shown.

Significance will be set at alpha = 0.05. Stata version 18 (StataCorp, College Station, TX) will be used for data analysis.

## 2.8. Data collection and management

**2.8.1. Data collection.**   Data collection will include age, gender, ANZICS APD number, admission diagnosis, ICU admission source, APACHE co-morbidities, admission diagnosis, and score.

Time measurements collected are ICU LOS, total duration of mechanical ventilation, total duration of vasoactive medication requirement, incidence and duration of renal replacement therapy and post-ICU ward LOS and day 60 mortality.

The laboratory data that will be recorded are pregnancy test in women of childbearing age, baseline hormones (sex hormone binding globulin, testosterone), routine blood tests (haemoglobin, white cell count, albumin, urea and creatinine, alanine transferase), lipids at baseline and at day 21. Inflammatory markers (CPR, procalcitonin and WBC count) will be recorded at enrolment, ICU discharge, and those as per routine care during the rest of the hospital stay.

Other measures collected are nutrition (caloric intake (kcal/total), protein intake (g/total)), hospital disposition (home vs other residential care vs rehabilitation), ICU readmission, hospital readmission, BMI, weight at hospital admission and pre-admission medications.

Data will be collected on a prespecified REDCap© electronic data which performs data entry validation. Data will be collected by trained research co-ordinators at each site. Patients and/or their next of kin will be asked to provide three possible points of contact (home and close family contact details) to the research staff prior to discharge. Day 90 & 180 (post hospital discharge) SF-36 will be conducted over the phone by a trained assessor.

**2.8.2. Data management and safety monitoring board.**   An independent Data Monitoring Committee (DMC), consisting of experts in intensive care and rehabilitation, will be established before patient enrolment to ensure safety of participants through monitoring of trial procedures, AE's and SAE's. No interim analysis is planned. However, all SAEs will be provided to the DSMB, and the DSMB reserves the right to call for a blinded or unblinded interim analysis at any point in the conduct of the trial.

Study data will be entered directly into a secure web-based case report, which will assign each patient a unique study number, without personal identifiers, preserving confidentiality. Once the study is completed, paper records will be shredded, and electronic database will be destroyed 15 years after the completion of the study. Only authorised study personnel will have access to the electronic database. Patients will be allocated a unique study number. The site research co-ordinator will compile an enrolment log that includes the patients' details and a unique study number. Study data will be obtained from the patients' medical records. The study data and the study enrolment logs will be kept separately. Contact details for participants and their next of kin will be provided to the project manager and quality of life outcome data will be obtained by a research coordinator by phone calls to participants and/or their next of kin. Following completion of the study, the data that support the findings of this study will be available from the corresponding author upon reasonable request.

## 2.9. Safety and adverse events

It is recognised that the intensive care patient population will experience a number of common aberrations in laboratory values, signs and symptoms due to the severity of the underlying disease and the impact of standard therapies. Intensive care patients will frequently develop life-threatening organ failure(s) unrelated to study interventions and despite optimal management. Therefore, consistent with established practice in academic ICU trials [31], events that are part

of the natural history of the primary disease process or expected complications of critical illness will not automatically be reported as serious adverse events in this study. All adverse events which are potentially causally related to the study intervention or are otherwise of concern in the investigator's judgement will be reported.

Potential side effects are cardiac failure, myocardial ischemia, virilisation in females (hoarseness, acne, hirsutism), abnormal LFTs, polycythaemia, and increased sensitivity to oral anticoagulants. Safety endpoints include new onset hypertension (SBP>180units), abnormal liver function tests (ALT increase > 100% from baseline OR >5 times normal limit), cardiac ischemia or heart failure or virilisation in females, new fall in HDL cholesterol, result <1mmol/L. On discharge to the ward, the accepting team will be informed that the patient has been part of the study and instructions will be given to the medical team to contact the principal investigator if the subject shows any of the potential side effects after ICU discharge.

### 2.10. Ethics and consent

The majority of patients enrolled in this trial will have capacity to give consent at the time of trial enrolment. The following consent options are acceptable: (i) consent by the patient (ii) consent by a substitute decision maker using the IMP / GAA pathway for WA Health; (iii) delayed consent from the patient. All participants who recover sufficiently will be given the opportunity to provide informed consent for ongoing study participation and for the use of data collected for the study. Ethics approval has been obtained from the Sir Charles Gairdner Hospital Human Research Ethics Committee in Western Australia. Auditing of clinical trial practices will occur under the governance of Sir Charles Gairdner Hospital Department of Research. Protocol amendments will be updated on relevant clinical trial registries by the Project Manager and communicated to the ethics committee for approval. Prospective trial registration has occurred with ANZCTR (number: ACTRN12623000729628).

## 3. Discussion

Functional recovery after critical illness is of great importance to patients and to the health care system. To our knowledge, this trial will be the largest trial evaluating nandrolone in intensive care patients to attempt to enhance recovery. The outcome measures chosen are patient-centred outcomes, and the intervention is relatively simple to provide. It will be combined with the usual rehabilitation efforts provided to recovering patients in the Intensive care, such as physiotherapy and nutrition.

The agent and dose selection required some consideration. The relatively higher myogenic potential of nandrolone over testosterone swayed this decision from a safety point of view. Nandrolone is currently approved in Australia and by the FDA for use in anaemia, CKD, osteoporosis, cachexia, and muscular dystrophy. In our pilot feasibility study nandrolone showed a reasonable safety profile with nil significant adverse events, however the results were limited by the small sample size (22 patients) [27]. Recruitment from this trial will come from a number of sites, which will help with the generalisability of the results. We look forward to providing the results to the critical care community once the trial is complete.

## Supporting information

**S1 Checklist. SPIRIT 2013 checklist: Recommended items to address in a clinical trial protocol and related documents*.**
(DOC)

**S1 File.**
(PDF)

## Author Contributions

**Conceptualization:** Matthew Anstey, Bradley Wibrow.

**Data curation:** Matthew Anstey, Maryam Habibi, Bianca Mammana, Annamaria Palermo, Xavier Fiorilla, Bhaumik Mevavala.

**Formal analysis:** Angela Jacques.

**Investigation:** Matthew Anstey, Ed Litton, Lisa Van der Lee, Robert Palmer, Natalie Tran, Bianca Mammana, Stacey Scheepers, Annamaria Palermo, Xavier Fiorilla, Bhaumik Mevavala, Adrian Regli, Bradley Wibrow.

**Methodology:** Matthew Anstey, Maryam Habibi, Stacey Scheepers, Angela Jacques, Bradley Wibrow.

**Resources:** Lisa Van der Lee, Robert Palmer.

**Supervision:** Matthew Anstey, Ed Litton, Bradley Wibrow.

**Writing – original draft:** Matthew Anstey, Maryam Habibi.

**Writing – review & editing:** Matthew Anstey, Ed Litton, Lisa Van der Lee, Robert Palmer, Natalie Tran, Bianca Mammana, Stacey Scheepers, Annamaria Palermo, Xavier Fiorilla, Bhaumik Mevavala, Adrian Regli, Angela Jacques, Bradley Wibrow.

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
