## [Decision Letter · Decision Letter 0]

17 Jul 2024

PONE-D-24-22777Muscle growth and anabolism in intensive care survivors (GAINS 2.0): Protocol for a multi-centre randomised; placebo controlled clinical trial of nandrolone in deconditioned adults recovering from critical illnessPLOS ONE

Dear Dr. Anstey,

Thank you for submitting your manuscript to PLOS ONE. After careful consideration, we feel that it has merit but does not fully meet PLOS ONE’s publication criteria as it currently stands. Therefore, we invite you to submit a revised version of the manuscript that addresses the points raised during the review process.

We look forward to receiving your revised manuscript.

Kind regards,

Francesco Sessa, Ph.D., MS

Academic Editor

PLOS ONE

Journal Requirements:

4. Please include your tables as part of your main manuscript and remove the individual files. Please note that supplementary tables (should remain/ be uploaded) as separate ""supporting information"" files"

Additional Editor Comments:

Reviewers have raised several concerns about the paper. Please address all the feedback provided, paying special attention to the criticisms from Reviewer#2.

Reviewers' comments:

Reviewer's Responses to Questions

**Comments to the Author**

1. Does the manuscript provide a valid rationale for the proposed study, with clearly identified and justified research questions?

Reviewer #1: Yes

Reviewer #2: Yes

2. Is the protocol technically sound and planned in a manner that will lead to a meaningful outcome and allow testing the stated hypotheses?

Reviewer #1: Partly

Reviewer #2: Partly

3. Is the methodology feasible and described in sufficient detail to allow the work to be replicable?

Reviewer #1: No

Reviewer #2: No

4. Have the authors described where all data underlying the findings will be made available when the study is complete?

Reviewer #1: Yes

Reviewer #2: No

5. Is the manuscript presented in an intelligible fashion and written in standard English?

Reviewer #1: Yes

Reviewer #2: Yes

6. Review Comments to the Author

You may also provide optional suggestions and comments to authors that they might find helpful in planning their study.

Reviewer #1: Dear Editor,

Thank you for the opportunity to review the study by Anstey et al.

Blinding Procedure: The blinding procedure must be more clearly described. It should be explicitly stated how placebo formulations and preparations are done in the study - what is used as placebo, what volume etc.. The details on how blinding will be maintained throughout the study should be expanded - i.e. that there are no information rute from the person preparing the study drug to the one administering the drug.

Randomization: The current randomization method involves the use of sealed envelopes. This approach is considered outdated compared to more modern solutions such as computer-generated randomization. The study should justify the choice of sealed envelopes. Clarify if any stratification or block randomization will be implemented to ensure balance among different patient groups.

Inclusion Criteria: The inclusion criteria need to be more specific. It is important to define the exact conditions under which patients will be assessed for inclusion. The current criteria suggest that any sedated or otherwise immobilized patient could be included. Clear guidelines should be established to determine which patients are eligible for the study to avoid ambiguity.

Power Calculation: The power calculation should be extended to include assumptions regarding the distribution of the outcome, supported by the best available data. The current mention of proportions in relation to a time-to-event outcome is unclear and needs clarification. Length of Stay (LOS) is a challenging outcome due to mortality bias. The study should justify why LOS was chosen over more recognized metrics such as days alive and out of hospital (DAOH) as the primary outcome.

Statistical Significance Level: The statistical significance level should be explicitly reported. Additionally, a power calculation for secondary outcomes should be provided to ensure that the study is adequately powered to detect differences in these outcomes as well.

Statistical Analysis: As this is a randomized study, it is generally discouraged to adjust for confounding factors. The study should consider not adjusting for these factors to maintain the integrity of the randomization process and avoid introducing potential biases.

Reviewer #2: This is a randomised controlled trial of nandrolone versus saline injections in patients from the ICU. Doses vary by sex.

There are only 60 patients to be recruited. The randomisation is stratified by site - which is okay so long as blinding is sufficiently good, but given the different dose by sex, stratification by sex is important too.

The sample size calculation is entirely inadequate here. Please give the actual figures used to derive a total number of patients.

The analyst cannot be truly blind to assignment - this needs to be reworded.

How are people who do not leave hospital - or die in hospital - allowed for in the length of stay analysis. Length of stay is unlikely to be normally distributed. There are unlikely to be enough patients and events to allow for the amount of adjustment mentioned here.

The people referenced in the text are Student (capital S) and R A Fisher (not Fischer). Why are both logrank tests and Cox regression being used for unadjusted analyses? Which RR is going to be the primary analysis?

Has the acceptability of a dummy IM injection - which comes with issues of discomfort etc - been piloted within the hospital population of interest?

7. PLOS authors have the option to publish the peer review history of their article (what does this mean?). If published, this will include your full peer review and any attached files.

Reviewer #1: No

Reviewer #2: No

---

## [Decision Letter · Decision Letter 1]

17 Sep 2024

PONE-D-24-22777R1Muscle growth and anabolism in intensive care survivors (GAINS 2.0): Protocol for a multi-centre randomised; placebo controlled clinical trial of nandrolone in deconditioned adults recovering from critical illnessPLOS ONE

Dear Dr. Anstey,

Thank you for submitting your manuscript to PLOS ONE. After careful consideration, we feel that it has merit but does not fully meet PLOS ONE’s publication criteria as it currently stands. Therefore, we invite you to submit a revised version of the manuscript that addresses the points raised during the review process.

We look forward to receiving your revised manuscript.

Kind regards,

Francesco Sessa, Ph.D., MS

Academic Editor

PLOS ONE

Journal Requirements:

Additional Editor Comments:

Following the reviewers' comments, the authors have improved their manuscript. I believe the paper can be published with a few additional modifications. Please provide justifications for any further reviewers' comments.

Reviewers' comments:

Reviewer's Responses to Questions

**Comments to the Author**

1. Does the manuscript provide a valid rationale for the proposed study, with clearly identified and justified research questions?

Reviewer #1: Yes

Reviewer #2: Yes

2. Is the protocol technically sound and planned in a manner that will lead to a meaningful outcome and allow testing the stated hypotheses?

Reviewer #1: Yes

Reviewer #2: Yes

3. Is the methodology feasible and described in sufficient detail to allow the work to be replicable?

Reviewer #1: Yes

Reviewer #2: Yes

4. Have the authors described where all data underlying the findings will be made available when the study is complete?

Reviewer #1: No

Reviewer #2: No

5. Is the manuscript presented in an intelligible fashion and written in standard English?

Reviewer #1: Yes

Reviewer #2: Yes

6. Review Comments to the Author

You may also provide optional suggestions and comments to authors that they might find helpful in planning their study.

Reviewer #1: Dear Editor,

The manuscript has been improved on several key aspects. There are however still areas with ambiguity and areas that need clarification.

It is unclear whether the adjusted mixed effects models are intended as the primary or secondary/sensitivity analysis. If they are intended as the primary analysis, this should be strongly justified and factored into the sample size calculation. There are 6 variables for 60 patients, which may lead to an inflation of statistical power. In general, adjustment is not recommended in the primary analysis of randomized trials.

The "days out of hospital" outcome is not mentioned or defined in the outcomes section. Please also explain how patients who die during follow-up are handled. How is missing follow-up data addressed?

In the sample size section, the authors justify needing approximately 45 patients but round this number up to 60 due to an expected 25% loss to follow-up. A 25% expected loss to follow-up is a significant number that could skew the results. The authors must justify this estimate and explain how it will be addressed, including any potential bias it may introduce. If the 25% figure is more of a convenience decision to reach a sample size of 60, please consider revising to only include the number necessary to avoid exposing additional patients to the intervention unnecessarily.

Given the complexity of the described analysis, it is recommended to complete a separate, full statistical analysis plan before unblinding the data.

Reviewer #2: Thank you for your response to my preivous concenrs - I do not see a data sharing protocol here and it should be in the data management section here.

7. PLOS authors have the option to publish the peer review history of their article (what does this mean?). If published, this will include your full peer review and any attached files.

Reviewer #1: No

Reviewer #2: No

---

## [Author Response · Author response to Decision Letter 1]

9 Oct 2024

Thank you to the Editor and Reviewers for your thoughtful comments on the trial paper. It was helpful to refine the paper.

Our responses are outlined below.

Reviewer #1: 

It is unclear whether the adjusted mixed effects models are intended as the primary or secondary/sensitivity analysis. If they are intended as the primary analysis, this should be strongly justified and factored into the sample size calculation. There are 6 variables for 60 patients, which may lead to an inflation of statistical power. In general, adjustment is not recommended in the primary analysis of randomized trials.

Thank you, primary outcome models will not be adjusted.

The "days out of hospital" outcome is not mentioned or defined in the outcomes section. Please also explain how patients who die during follow-up are handled. How is missing follow-up data addressed?

Days out of hospital will be modelled using survival methods if date of death is available,censoring on post-discharge death. If date of death not available ,number of days for non-survivors set to zero.

In the sample size section, the authors justify needing approximately 45 patients but round this number up to 60 due to an expected 25% loss to follow-up. A 25% expected loss to follow-up is a significant number that could skew the results. The authors must justify this estimate and explain how it will be addressed, including any potential bias it may introduce. If the 25% figure is more of a convenience decision to reach a sample size of 60, please consider revising to only include the number necessary to avoid exposing additional patients to the intervention unnecessarily.

Thank you. In using 25%, we considered the patient cohort, ie those recovering from severe critical illness, and that is what we originally based the 25% loss to follow-up on. We have re-looked at our pilot study, which had 2 deaths out of 22 patients – so we have revised the sample size figure to a 15% loss to follow up = 54 patients.

Given the complexity of the described analysis, it is recommended to complete a separate, full statistical analysis plan before unblinding the data.

A statistical analysis plan will be prepared and submitted to IDMC if required, for approval prior to unblinding.

Reviewer #2: 

Thank you for your response to my previous concerns - I do not see a data sharing protocol here and it should be in the data management section here.

Thank you. In the PLOSONE template we have the following statement : No datasets were generated or analysed during the current study. All relevant data from this study will be made available upon study completion. 

We have added this sentence “Following completion of the study, the data that support the findings of this study will be available from the corresponding author upon reasonable request.”

---

## [Decision Letter · Decision Letter 2]

22 Nov 2024

Muscle growth and anabolism in intensive care survivors (GAINS 2.0): Protocol for a multi-centre randomised; placebo controlled clinical trial of nandrolone in deconditioned adults recovering from critical illness

PONE-D-24-22777R2

Dear Dr. Anstey,

We’re pleased to inform you that your manuscript has been judged scientifically suitable for publication and will be formally accepted for publication once it meets all outstanding technical requirements.

Kind regards,

Francesco Sessa, Ph.D., MS

Academic Editor

PLOS ONE

Additional Editor Comments (optional):

Following the reviewers' suggestions, the authors iimproved their manuscript. I endorse the publication in its current form.

Reviewers' comments:

Reviewer's Responses to Questions

**Comments to the Author**

1. Does the manuscript provide a valid rationale for the proposed study, with clearly identified and justified research questions?

Reviewer #2: Yes

2. Is the protocol technically sound and planned in a manner that will lead to a meaningful outcome and allow testing the stated hypotheses?

Reviewer #2: Yes

3. Is the methodology feasible and described in sufficient detail to allow the work to be replicable?

Reviewer #2: Yes

4. Have the authors described where all data underlying the findings will be made available when the study is complete?

Reviewer #2: Yes

5. Is the manuscript presented in an intelligible fashion and written in standard English?

Reviewer #2: Yes

6. Review Comments to the Author

You may also provide optional suggestions and comments to authors that they might find helpful in planning their study.

Reviewer #2: Thank you for responding to my previous comments - I have nothing more to add and am happy to pass thuis for publciation

7. PLOS authors have the option to publish the peer review history of their article (what does this mean?). If published, this will include your full peer review and any attached files.

Reviewer #2: No

---

## [Editor Report · Acceptance letter]

11 Dec 2024

PONE-D-24-22777R2 

PLOS ONE

Dear Dr. Anstey, 

I'm pleased to inform you that your manuscript has been deemed suitable for publication in PLOS ONE. Congratulations! Your manuscript is now being handed over to our production team.

Kind regards, 

on behalf of

Lecturer Francesco Sessa 

Academic Editor

PLOS ONE